# Efficient transfer learning for NLP with ELECTRA

## Reproducibility Summary

**Scope of Reproducibility**

Clark et al. [2020] claims that the ELECTRA approach is highly efficient in NLP performances relative to computation budget. As such, this study focus on this claim, summarized by the following question: *Can we use ELECTRA to achieve close to SOTA performances for NLP in low-resource settings, in term of compute cost?*

**Methodology**

This replication study has been conducted by fully reimplementing the small variant of the original ELECTRA model (Clark et al. [2020]). All experiments are performed on single GPU computers. GLUE benchmark dev set (Wang et al. [2018]) is used for models evaluation and compared with the original paper.

**Results**

My results are similar to the original ELECTRA's implementation (Clark et al. [2020]), despite minor differences compared to the original paper for both implementations. With only 14M parameters, ELECTRA-Small outperforms, in absolute performances, concurrent pretraining approaches from some previous SOTA, such as GPT, or alternative efficient approaches using knowledge distillation, such as DistilBERT. By taking into account compute cost, ELECTRA is clearly outperforming all compared approaches, including BERT and TinyBERT. **Therefore, this work supports the claim that ELECTRA achieves high level of performances in low-resource settings, in term of compute cost.**

Furthermore, with an increased generator capacity than recommended by Clark et al. [2020], the discriminant can collapses by being unable to distinguish if inputs are fake or not. Thus, while ELECTRA is easier to train than GAN (Goodfellow et al. [2014]), it appears to be sensitive to capacity allocation between generator and discriminator.

The code and a pretrained model will be released.

**What was easy**

Information provided by the authors of the original paper (Clark et al. [2020]), either from the paper; within the source code: or from the official Github repository, is very rich and exhaustive to understand the proposed approach. In addition, as stated with their main claim, ELECTRA can be easily run on a single GPU.

**What was difficult**

By being an aggregation of several tasks and the variance from results, GLUE benchmark requires significant amount of effort, in term of implementation and computation. For models comparison, several tricks can also influence the results, which is even more amplified by the different aggregation formulas and the lack of measure of dispersion for published results. As such, confirming the correctness of this reimplementation was harder than expected.

**Communication with original authors**

Kevin Clark, one of the original authors, has been helpful by answering some questions. Unfortunately, breakdown of GLUE score per tasks have not yet been provided to fully compare this implementation with the original one. Otherwise, most of questions that I had were already answered though the Github repository or by inspecting the source code.

Submitted to ML Reproducibility Challenge 2020. Do not distribute.

# 1  Introduction

Over the recent years, the combination of pretraining task with large unlabelled text corpus has shown a lot of success for Natural Language Processing (NLP) tasks (Howard and Ruder [2018], Peters et al. [2018], Devlin et al. [2019], Liu et al. [2019], Radford et al. [2019]). However, the main drawback is the computation requirements, which limits the adoption of these approaches for more specialized domains for some potential use cases.

Unlike the recent trend of increasing computing requirements, in the original paper ELECTRA (Clark et al. [2020]), the authors are taking the opposite research direction to have more efficient approach for transfer learning for NLP tasks. Thus, the authors provide a way to democratize the use of deep learning for more actors and more use cases.

# 2  Scope of reproducibility

The scope for this reproducibility work is related to the following questions, which are part of the original paper (Clark et al. [2020]).

***Can we use ELECTRA pretraining task to achieve close to SOTA performances on low-resource settings, in term of compute cost?***
To formalize this claim, we use the GLUE benchmark (Wang et al. [2018]) as measure of NLP performances and compare it with large pretrained models, such as GPT (Radford et al. [2018]). All experiments are executed on single GPU computers.

***How does the training process behave with different generator size ?***
While ELECTRA shares some aspects from GAN (Goodfellow et al. [2014]), with the use of discriminator and generator networks, the training is not adversarial. Nevertheless, the capacity of the generator compared to the discriminator remains an important hyper-parameter in the architecture.

# 3  Methodology

The approach for this work is to fully reimplement in PyTorch the small variant of ELECTRA (Clark et al. [2020]). For this, the different libraries from HuggingFace (Wolf et al. [2020]), namely Transformers; Tokenizers; and Datasets, have been used respectively for the Transformer (Vaswani et al. [2017]) implementation and training loop logic; tokenization models; and to ease the access to relevant datasets.

Due to constraint of compute resources availability initially, some changes have been implemented in this reimplementation, such as the ability to use gradient accumulation. Furthermore, to ease further explorations, as the preprocessing steps are time consuming, approximately 3 hours, the reimplementation's preprocessing has been adapted to be less dependant on the sequence length, see section 3.2 for the details.

## 3.1  Model descriptions

ELECTRA-Small is a complete reimplementation in PyTorch of ELECTRA-Small*, original implementation in TensorFlow from Clark et al. [2020]. The architecture consists in two Transformer based neural networks:

- The generator network transforms a sequence of tokens, containing masked tokens, into a new sequence with the original tokens predictions for these masked tokens.

- The discriminator network transforms a sequence of tokens, containing tokens replaced by a generator, into a new sequence of binary predictions, true if the token has been replaced, otherwise false.

Similar to ELECTRA-Small* (Clark et al. [2020]), input embeddings are shared between the generator and the discriminator models. Moreover, input token embeddings and output token embeddings are shared for the generator model. Position embeddings are also used similar to BERT (Devlin et al. [2019]) in both ELECTRA implementations.

For the downstream tasks only, both implementations use a 2 layer MLP heads. The original implementation (Clark et al. [2020]) uses the first contextualized embeddings for the downstream head. However, this reimplementation uses average pooling across all contextualized embeddings.

For a visual representation, please refer to the figure 1.

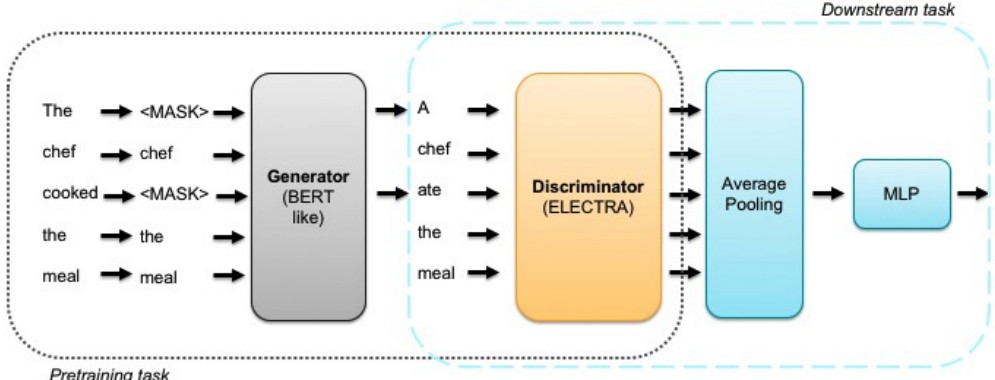

Figure 1: Model architecture.
Generator is only used at pretraining. Average pooling and MLP are only used for downstream tasks.
Note: the original implementation (Clark et al. [2020]) uses first embedding for pooling layer instead of average pooling.
Image inspired by the original paper (Clark et al. [2020]).

## 3.2 Datasets

**Pretraining task:** The original paper (Clark et al. [2020]) uses the same datasets and preprocessing steps as BERT (Devlin et al. [2018]), namely Wikipedia and BookCorpus (Zhu et al. [2015]), also referred as Wikibooks. In the official GitHub repository[1], the authors use also OpenWebText, (Gokaslan and Cohen [2019]), referred as OWT. The preprocessing steps consist to extract text segments satisfying the maximum sequence length from large text corpus. Segments are then tokenized using WordPiece (Wu et al. [2016]) with a vocabulary size 30,522. For each training input, two tokenized segments are concatenated with the special token <SEP> and the special token <CLS> is added in the first position. In addition of this input, a sequence id is provided to identify the segment, either 0 or 1, and a position id is also provided to give spacial information into inputs.

For this reproducibility study, the dataset for the pretraining task is OpenWebText (Gokaslan and Cohen [2019]). For the preprocessing, the reimplementation uses a slight variation from the original paper. Indeed, all documents are tokenized offline using Byte-level Byte Pair Encoding (Radford et al. [2019]) with a vocabulary size 30,522. During training, for each document, a random segment with the maximum sequence length allowed by the model is selected. In other words, this reimplementation uses a dynamic segmentation instead of precomputed static segments in the original implementation.

**Downstream task:** Similar to the original paper (Clark et al. [2020]), the downstream tasks are the GLUE benchmark (Wang et al. [2018]). The preprocessing steps are mostly similar to the original paper. For single sentence tasks, inputs consist of the single sentence for each training input. For double sentences tasks, both sentences are concatenated with the special token <SEP>. Finally, inputs are potentially truncated to ensure the maximum sequence length.

## 3.3 Hyperparameters

The selected hyperparameters are the same as the original paper, except for the optimization algorithm and the use of gradient accumulation. For the optimizer, AdamW (Loshchilov and Hutter [2019]) has been used instead of Adam (Kingma and Ba [2014]). In addition, the original implementation is actually using a lower layer-wise learning rate than described in Clark et al. [2020].

For more information, please refer to the tables 5 and 6.

## 3.4 Experimental setup

Pretrained models were trained on a single GPU computer, with Nvidia RTX 3090. Regarding the downstream tasks, experiments were either run on Google Colab Pro instances, with Nvidia P100 or V100. Due to the heterogeneous

---

[1]https://github.com/google-research/electra

Table 1: Compute costs for the different experiments in GPU.days, relative to Nvidia RTX 3090. The overall total is less than the sum of individual experiments due to overlap.

| Experiment | Task | Run cost | # config. | # seeds | Total |
|---|---|---|---|---|---|
| Reproducing ELECTRA on GLUE benchmark (section 4.1) | Pretraining | 5 | 1 | 1 | 5 |
|  | Downstream | 0.2 | 6 | 5 | 6 |
| Sensitivity to generator size (section 4.2) | Pretraining | $25\% \times 5$ | 4 | 1 | 5 |
|  | Downstream | 0.2 | 4 | 5 | 4 |
| Overall |  |  |  |  | 17.5 |
| *Overall USD cost assuming 1 V100 GPU at USD2.26 hourly rate* |  |  |  |  | *$962* |

infrastructure, a key element of the experimental setup is the centralization of results with Weights & Biases (Biewald [2020]). For further details, please refer to the Github repository[2].

## 3.5 Computational requirements

The pretraining task and the downstream tasks can be performed with a relative limited GPU memory budget and a single GPU, thanks to gradient accumulation and mixed precision. For example, pretraining can be done with a single GTX 1060 6GB in 21 days; or on GTX 1080 Ti 11GB in 13 days instead of 3.75 days on RTX 3090 24GB.

Interestingly, by constraining the GPU memory usages to 16GB, minimum amount for the GPU used in the original paper (Clark et al. [2020]), the duration for the pretraining task is similar to the original implementation (Clark et al. [2020]), meaning 3.75 days on RTX 3090 instead of 4 days on V100. Using all GPU memory for the RTX 3090 would have decrease the computation requirements, but it would have complicated the comparison between reimplementations.

Computational requirements, per experiments and overall, are summarized in the table 1.

## 4 Experiments

### 4.1 Reproducing ELECTRA on GLUE benchmark

In this experiment, one model is pretrained with 1 million steps on OpenWebText (Gokaslan and Cohen [2019]) with the similar approach as the original paper (Clark et al. [2020]). With the weights initialization from this pretrained model, the downstream task, the GLUE benchmark (Wang et al. [2018]), is executed 5 times with different seeds. The results are summarized in the table 2.

**Reimplementation compared to the original:** This reimplementation provides results, which are fairly similar to the original implementation (Clark et al. [2020]) on OpenWebText dataset for individual GLUE tasks. Furthermore, the training dynamics, captured by the different metrics from the generator and the discriminator, are very similar to the one provided in the official ELECTRA's Github repository, see figure 3 in appendix. Thus, this reimplementation behaves similarly to the original implementation.

**Discrepancies for the aggregated GLUE score:** By comparing results in table 2 and table 1 from Clark et al. [2020], this study reports lower GLUE scores for each baseline and ELECTRA models. For baseline models, I collected the different information from different sources, mainly from the original papers (see notes from table 2 for exact sources). Then, I recomputed the GLUE score on dev set as per Wang et al. [2018] from the individual task results. The discrepancy appears to be related to a difference of reporting methodology, surely related to the treatment of the WNLI task. Therefore, table 1 from Clark et al. [2020] is not reporting the actual GLUE score on dev set but a meta score, from which I cannot fully reproduce. While this discrepancy involves an overstatement for all models in Clark et al. [2020], it doesn't change the main claim about the high performance of ELECTRA relative to other approaches.

**Robustness:** Despite the reported bug[3] from the original implementation regarding layer-wise learning rate decay; the specific data augmentation procedure for STS and MRPC task[4], which is not used in this reimplementation; and the minor differences in preprocessing steps, see section 3.2, the results are very close between this reimplementation and

---

[2]tobeaddedlater

[3]https://github.com/google-research/electra/issues/51

[4]https://github.com/google-research/electra/issues/98

Table 2: Results on GLUE dev set. For this reimplementation, results are reported as mean and standard deviation, using 5 different seeds per task.
(*mc*: Matthews correlation, *acc*: Accuracy, *spc*: Spearman correlation, AVG: Average of individual metrics as Clark et al. [2020])

| Model | CoLA (mc) | SST-2 (acc) | MRPC (acc) | STS-B (spc) | QQP (acc) | MNLI (acc) | QNLI (acc) | RTE (acc) | AVG | GLUE* |
|---|---|---|---|---|---|---|---|---|---|---|
| **Baselines (≫14M parameters)** | | | | | | | | | | |
| ELMo[2] | 15.6 | 84.9 | 80.6 | 64.4 | 82.2 | 69.4 | 73.0 | 50.9 | 65.1 | 63.8 |
| ELMo frozen[1] | 44.1 | 91.5 | 82.3 | 70.5 | 84.3 | 68.6 | 71.2 | 53.4 | 70.7 | 68.7 |
| GPT[2] | 50.2 | 93.2 | 85.9 | 86.5 | 85.9 | 81.2 | 82.4 | 58.1 | 77.9 | 75.4 |
| DistilBERT$_6$[3] | 51.3 | 91.3 | 87.5[†] | 86.9[†] | 88.5[†] | 82.2 | 89.2 | 59.9 | | 77.0 |
| TinyBERT$_6$[4] | 54.0 | 93.0 | 86.3 | 89.6 | 91.1 | 84.5 | 91.1 | 73.4 | 82.9 | 80.0 |
| BERT-Base[3] | 56.3 | 92.7 | 88.6[†] | 89.0[†] | 89.6[†] | 86.7 | 91.8 | 69.3 | | 80.0 |
| **ELECTRA-Small OWT - Original (≈14M parameters)** | | | | | | | | | | |
| 100% trained[5] | 56.8 | 88.3 | 87.4 | 86.8 | 88.3 | 78.9 | 87.9 | 68.5 | 80.4 | |
| **ELECTRA-Small OWT - Mine (≈14M parameters)** | | | | | | | | | | |
| 6% trained | 44.4 ± 1.87 | 81.9 ± 0.83 | 83.4 ± 1.11 | 80.2 ± 0.37 | 83.6 ± 0.16 | 74.9 ± 0.21 | 84.1 ± 0.27 | 57.6 ± 2.76 | 74.3 ± 0.64 | 72.3 ± 0.61 |
| 12% trained | 46.1 ± 1.34 | 84.2 ± 0.46 | 83.0 ± 1.75 | 82.0 ± 0.35 | 84.2 ± 0.08 | 76.2 ± 0.12 | 84.7 ± 0.37 | 59.4 ± 2.61 | 75.4 ± 0.53 | 73.4 ± 0.45 |
| 25% trained | 50.2 ± 1.38 | 86.6 ± 0.66 | 85.3 ± 0.74 | 83.9 ± 0.61 | 85.0 ± 0.09 | 77.9 ± 0.19 | 86.0 ± 0.14 | 58.2 ± 2.52 | 77.1 ± 0.33 | 74.8 ± 0.29 |
| 50% trained | 51.5 ± 0.98 | 88.0 ± 0.66 | 86.7 ± 1.47 | 84.3 ± 0.60 | 85.5 ± 0.08 | 79.2 ± 0.23 | 86.8 ± 0.36 | 60.8 ± 1.54 | 78.3 ± 0.40 | 75.9 ± 0.34 |
| 75% trained | 53.8 ± 1.52 | 89.2 ± 0.67 | 87.0 ± 0.47 | 84.8 ± 0.47 | 85.9 ± 0.08 | 79.8 ± 0.11 | 87.2 ± 0.50 | 61.6 ± 1.23 | 79.1 ± 0.26 | 76.6 ± 0.24 |
| 100% trained | 53.5 ± 2.47 | 88.7 ± 0.53 | 87.6 ± 1.58 | 85.2 ± 0.36 | 86.1 ± 0.18 | 80.2 ± 0.13 | 87.5 ± 0.34 | 61.5 ± 0.97 | 79.2 ± 0.30 | 76.7 ± 0.25 |

[1] Figures from Wang et al. [2018].
[2] Figures from Phang et al. [2019].
[3] Figures from Sanh et al. [2020].
[4] Figures from Jiao et al. [2020].
[5] Figures from the official ELECTRA's Github repository. The associated GLUE score cannot be computed as F1 scores and Person correlation are unavailable.
[†] Figures are the average of the 2 metrics (F1 and accuracy for MRPC and QQP; and Person and Spearman correlation for STS) and are not comparable.
[*] GLUE scores are recomputed with the required metrics for each GLUE tasks. For WNLI. as the majority class predictor beats all models, the GLUE scores use 56.34% accuracy for this specific task.

the original one (Clark et al. [2020]). Thus, we can conclude the ELECTRA approach is robust and generalizable as results are similar despite different implementations and different datasets (Wikibooks and OWT).

**Efficiency:** I also compared results from this reimplementation to the literature with the hardware requirements. Findings are summarized in table 3. By looking at the computation requirements, estimated by heuristics (see OpenAI blog on AI and Compute), relative to GLUE scores, ELECTRA (Clark et al. [2020]) is outperforming all models. Interestingly, this observation is also valid for models using knowledge distillation such as DistilBERT (Sanh et al. [2020]) or TinyBERT (Jiao et al. [2020], even if we don't take into account the initial cost for the teacher model. A similar conclusion can be made by looking only at number of parameters, which is somehow an approximate proxy for inference time. Therefore, ELECTRA is outperforming all compared approaches in term of efficiency.

Table 3: Efficiency metrics for results on GLUE dev set. Peta-flops are estimated using heuristic from theoritical FLOPS per GPU; number of GPUs; and their utilization rate (see OpenAI blog on AI and Compute). Lower pfs-days is better. The compute cost is overstated for RTX 3090 due to the constraint to use only 16GB of GPU memory instead of the full 24GB.
(Pfs-days: peta-flops day; AVG: Average of individual metrics as ELECTRA (Clark et al. [2020])

| Model | Parameters | Train time + hardware | Pfs-days[†] | AVG[‡] | GLUE[‡] | Pfs-day per AVG % | Pfs-day per GLUE % |
|---|---|---|---|---|---|---|---|
| Baselines | | | | | | | |
| ELMo | 93M[a] | 14d on 3 GTX 1080[b] | ≈ 0.12 | 65.1 | 63.8 | ≈ 0.19 (3.5x) | ≈ 0.19 (3.4x) |
| ELMo frozen | 93M[a] | 14d on 3 GTX 1080[b] | ≈ 0.12 | 70.7 | 68.7 | ≈ 0.17 (3.2x) | ≈ 0.18 (3.2x) |
| GPT | 110M[c] | 30d on 8 P600[d] | ≈ 0.95 | 77.9 | 75.4 | ≈ 1.22 (22x) | ≈ 1.22 (22x) |
| DistilBERT$_6$ | 67M[e] | 90h on 8 V100[f] | ≈ 0.16 | | 77.0 | | ≈ 0.21 (3.6x) |
| TinyBERT$_6$ | 67M[e] | | | 82.9 | 80.0 | | |
| BERT-Base | 110M[g] | 4d on 16 TPU[g] | ≈ 0.95 | | 80.0 | | ≈ 1.19 (21x) |
| ELECTRA-Small OWT - Original | | | | | | | |
| 100% trained | 14M[h] | 4d on 1 V100[h] | ≈ 0.02 | 80.4 | | ≈ 0.03 (0.5x) | |
| ELECTRA-Small OWT - Mine | | | | | | | |
| 100% trained | 14M | 3.75d on 1 RTX 3090 | ≈ 0.04 | 79.2 | 76.7 | ≈ 0.05 (1.0x) | ≈ 0.06 (1.0x) |

[a] Information from Allen AI website .
[b] Information from ELMo github repository .
[c] Information from HuggingFace.
[d] Information from OpenAI blog on AI and Compute.
[e] Information from Jiao et al. [2020].
[f] Information from Sanh et al. [2020].
[g] Information from Devlin et al. [2019].
[h] Information from Clark et al. [2020].
[†] Formula from OpenAI blog on AI and Compute.
*Assumptions: 8.9TFLOPS GTX 1080; 12TFLOPS for P600; 16TFLOPS for V100; 45TFLOPS for TPU; 35TFLOPS for RTX 3090; 0.33 utilization rate.*
[‡] See table 2.

## 4.2 Sensitivity to generator size

In this experiment, different models, with different generator capacity and with the same discriminator capacity, are pretrained like in section 4.1 with that the exception the training was stopped at 250k steps (25%). The results are summarized in the table 4.

**Training stability:** Interestingly, in case of larger generator model than recommended in the original paper (Clark et al. [2020]), in this reimplementation, the discriminator network can collapse by not being able to distinguish which tokens are fake or not. Therefore, even if ELECTRA is not using an adversarial setting (Goodfellow et al. [2014]), the training procedure may collapse if we allocate too much capacity to the generator compared to the discriminator. Please refer to figure 2 for a visual representation.

# 5 Discussion

## 5.1 What was easy

**Ability to run on small GPU:** One claim of ELECTRA (Clark et al. [2020]) is the relative low computation requirements. In this work, only runs with a single GPU have been executed. Furthermore, the GPU memory requirement, roughly 15GB, can be easily decreased with the use of gradient accumulation. As example, with 2 gradient accumulation steps, the pretraining can be performed on a 11GB GPU.

Table 4: Results on GLUE dev set for different generator capacities trained with 250k steps. Results are reported as mean and standard deviation, using 5 different seeds per task. Gen.Size represents the multiplier for hidden-size, FFN-size and num-attention-heads for the generator network compared to the discriminator network.
(*mc*: Matthews correlation, *acc*: Accuracy, *spc*: Spearman correlation, AVG: Average of individual metrics as Clark et al. [2020])

| Model | CoLA (mc) | SST-2 (acc) | MRPC (acc) | STS-B (spc) | QQP (acc) | MNLI (acc) | QNLI (acc) | RTE (acc) | AVG | GLUE |
|---|---|---|---|---|---|---|---|---|---|---|
| ELECTRA-Small | | | | | | | | | | |
| 12.5% Gen.Size | 35.5 ± 3.51 | 85.9 ± 0.58 | 77.6 ± 0.75 | 82.0 ± 0.37 | 83.7 ± 0.12 | 76.1 ± 0.17 | 84.9 ± 0.45 | 55.0 ± 4.65 | 73.1 ± 0.90 | 71.4 ± 0.80 |
| 25% Gen.Size | 50.2 ± 1.38 | 86.6 ± 0.66 | 85.3 ± 0.74 | 83.9 ± 0.61 | 85.0 ± 0.09 | 77.9 ± 0.19 | 86.0 ± 0.14 | 58.2 ± 2.52 | 77.1 ± 0.33 | 74.8 ± 0.29 |
| 50% Gen.Size | Discriminator collapses at 10K steps | | | | | | | | | |
| 75% Gen.Size | Discriminator collapses at 10K steps | | | | | | | | | |
| 100% Gen.Size | Discriminator collapses at 10K steps | | | | | | | | | |

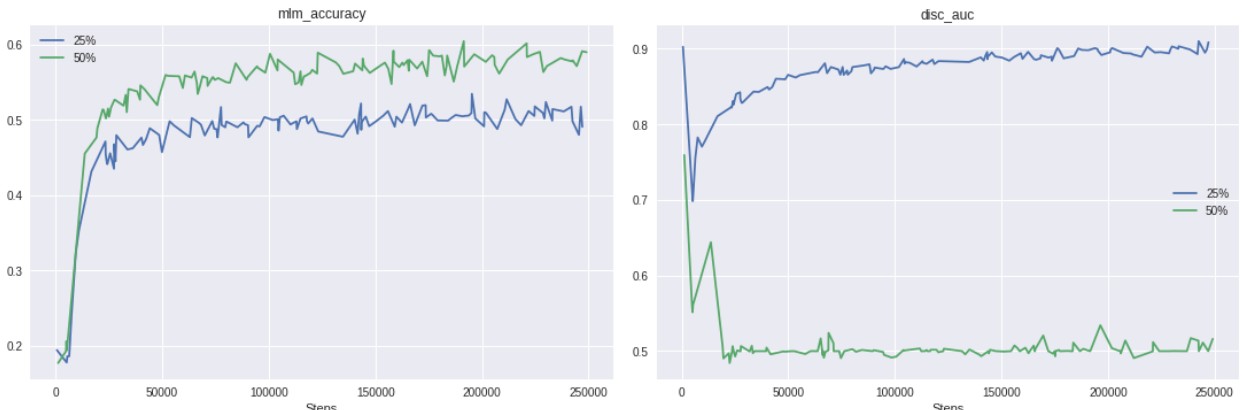

Figure 2: Generator (left for the accuracy metric) and discriminator (right for the AUC metric) behaviours for different generator size relative to the discriminator. For larger generator size than 25%, the discriminator may be unable to distinguish if inputs are fake or not.

**Clarity of the original paper:** The original paper (Clark et al. [2020]) contains a lot of detailed information, including the hyperparameters used for the pretraining and downstream tasks. The proposed pretraining task is also well explained and the replication was fairly easy as such. The only exception was the unavailability of all GLUE metrics, such as F1 score or Person correlation. The section about negative results is also very beneficial to better understand some early explorations which has lead the Authors to propose the ELECTRA approach. It would be beneficial for all if such openness about negative results would be more systematically present in research papers, along the positive results.

**Original source code and documentation:** Authors of the original paper (Clark et al. [2020]) have released their source code in their Github repository[5]. While the source code was developed in Tensorflow instead of the target framework, PyTorch, for this reimplementation, the original source code was relatively easy to understand. Plenty of documentation and answers were already present in this Github repository.

## 5.2 What was difficult

**GLUE benchmark:** The GLUE benchmark is composed on 9 different tasks. While each task is fairly simple to implement, implementing all of them to use the same pipeline requires some software engineering effort, at least further than expected. More importantly for comparison, results are subject to high variance and therefore, most papers use some statistics over several runs, usually mean or median. The multitude of runs increase significantly the computation budget. It would also be beneficial to also report the variance of the results to have a better sense of this variance, and

---

[5]https://github.com/google-research/electra

to ease comparisons between models. Finally, papers use different way to report results, such as meta score like in Devlin et al. [2019], or different metrics for MRPC, STS or QQP, for example Clark et al. [2020] and Sanh et al. [2020]. More practically, the validation of this ELECTRA reimplemention, compared to the original one, was harder due these reporting differences.

**Additional tricks for GLUE benchmark:** Submissions for the GLUE benchmark have been using additional tricks to improve performances, such as ensembling, further pretraining, multi-task training vs single task training, different hyperparameters between tasks. For example, while this has been mentioned in the original paper, see appendix from Clark et al. [2020], one specific task data augmentation procedure is also used for the results submissions for MRPC and STSB tasks. All these tricks make harder to compare different models, as described in Aßenmacher and Heumann [2020].

### Communication with original authors

Kevin Clark, one of the original authors (Clark et al. [2020]), has been helpful by answering some questions. Unfortunately, breakdown of GLUE score per tasks have not yet been provided to fully compare this implementation with the original one. Otherwise, most of questions that I had were already answered though the Github repository or by inspecting the source code.

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

 **Appendix**

Table 5: Hyper-pameters for the pretraining task.

| Hyperparameter | ELECTRA-Small Original | ELECTRA-Small Mine |
|---|---|---|
| Underlying base architecture | Transformer | Transformer |
| Number of layers at token level | 12 | 12 |
| Number of layers at sentence level | N/A | N/A |
| Hidden size | 256 | 256 |
| FFN inner hidden size | 1024 | 1024 |
| Attention heads | 4 | 4 |
| Attention head size | 64 | 64 |
| Embedding size | 128 | 128 |
| Token embeddings | Yes | Yes |
| Position embeddings | Yes | Yes |
| Generator size (multiplier for hidden-size, FFN-size, and num-attention-heads) | 0.25 | 0.25 |
| Generator layer size (multiplier for number of layers) | 1.0 | 1.0 |
| Mask percent | 15 | 15 |
| Learning rate | 5e$-$4 | 5e$-$4 |
| Learning rate decay | Linear | Linear |
| Warmup steps | 10000 | 10000 |
| Optimizer | Adam | **AdamW** |
| Adam $\epsilon$ | 1e$-$6 | 1e$-$6 |
| Adam $\beta_1$ | 0.9 | 0.9 |
| Adam $\beta_2$ | 0.9999 | 0.9999 |
| Attention dropout | 0.1 | 0.1 |
| Dropout | 0.1 | 0.1 |
| Weight decay | 0.01 | 0.01 |
| Batch size | 128 | 128 |
| Gradient accumulation steps | 1 | *2 on 11GB* GPU 1 on 24GB GPU |
| Mixed precision | No | **Yes** |
| Dataset | Wikibooks | **OpenWebText** |
| Tokenizer | WordPiece | **BBPE** |
| Vocab size | 30522 | 30522 |
| Train steps | 1M | 1M |

Table 6: Hyper-parameters for the GLUE tasks.

| Hyperparameter | ELECTRA-Small Original | ELECTRA-Small Mine |
|---|---|---|
| Learning rate | 3e−4 | 3e−4 |
| Layerwise LR decay | **0.8**[6] | 0.8 |
| Learning rate decay | Linear | Linear |
| Warmup steps | 10000 | 10000 |
| Optimizer | Adam | **AdamW** |
| Adam $\epsilon$ | 1e−6 | 1e−6 |
| Adam $\beta_1$ | 0.9 | 0.9 |
| Adam $\beta_2$ | 0.9999 | 0.9999 |
| Attention dropout | 0.1 | 0.1 |
| Dropout | 0.1 | 0.1 |
| Weight Decay | 0 | 0 |
| Batch size | 32 | 32 |
| Gradient accumulation steps | 1 | 1 |
| Mixed precision | No | **Yes** |
| Epochs | 10 for RTE and STS 3 for other GLUE benchmarks | 10 for RTE and STS 3 for other GLUE benchmarks |
| Pooling | First contextualized embedding | **Average pooling** |

---

[6]The original implementation has got a bug which involves using a lower layerwise learning rate decay, see https://github.com/google-research/electra/issues/51

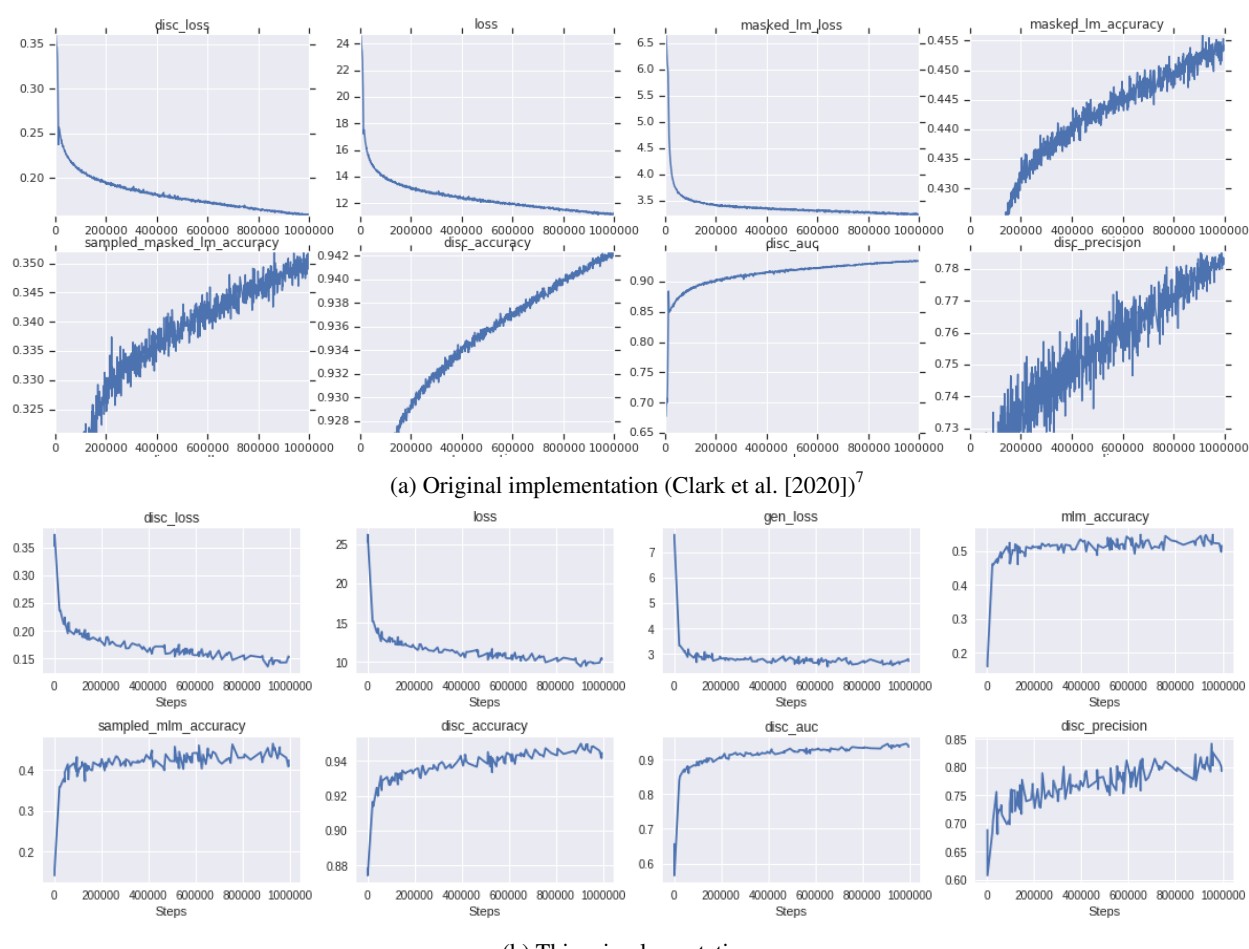

(a) Original implementation (Clark et al. [2020])[7]

(b) This reimplementation

Figure 3: Metrics and losses from generator and discriminator networks

---

[7]Figures from https://github.com/google-research/electra/issues/3

