# OpenReview forum: "Efficient transfer learning for NLP with ELECTRA"
_ML_Reproducibility_Challenge/2020 — Reject_

### Official Review · AnonReviewer1 · 2021-02-28

**Rating:** 6
**Confidence:** 5

**Review:**

# Reproducibility Summary

Yes it is provided.

# Scope of Reproducibility

Train ELECTRA with a similar dataset (OpenWebText) and evaluate on GLUE. Done a single GPU, this should reflect the claim of the ELECTRA paper of its effectiveness and relatively low computational cost to baselines such as BERT.

# Communication

The reproduction authors communicated with one of the original authors (partially through github) to answer most questions.

# Hyperparameter Search

The authors perform thorough hyperparameter search and it is reported clearly in the paper.

# Ablation Study

The authors perform some useful ablation with respect to discriminator and generator sizes. I think there is much to explore here such as other techniques that are used in GANs (e.g. different training schedules for the discriminator and generator).

# Discussion

It's clear from the discussion that ELECTRA is easy to reproduce. This seems attributed to the nature of the work — the ability to train on a single GPU — and the clarity and documentation associated with the original paper.


# Recommendations

No clear recommendation is made to the original authors, although the ablation experiments about generator size could be relevant.


# Results beyond the paper

The experiments and results are primarily in the spirit of the original work.

# Overall Organization and Clarity

The paper was clear and easy to follow. That being said, if needing to save space then some figures and tables could move to the appendix with only the main results in the main text.


**Familiar With The Original Paper:**

I have read the original paper

**Reproducibility Summary:**

Report has summary

---

### Official Review · AnonReviewer3 · 2021-03-01
**ELECTRA can be reproduced in PyTorch fairly well. But the results are sensitive to capacity allocation between generator and discriminator.**

**Rating:** 7
**Confidence:** 3

**Review:**

The reproducibility report shows ELECTRA can be reproduced quite well. The report is detailed and coherent. An additional insight is provided which is “sensitive to capacity allocation between generator and discriminator” by the authors. Finally, figure 3 is great.

The code had to be shared with the reviewers.

There are some adjustments in this re-implementation. This is both a risk to the reproducibility as the difference increases and a strength to demonstrate the core idea of ELECTRA works.

“can also influenced the” -> can also influence the


**Familiar With The Original Paper:**

I have not read the original paper

**Reproducibility Summary:**

Report has summary

---

### Official Review · AnonReviewer2 · 2021-03-03
**Clearly written paper with good replication results**

**Rating:** 9
**Confidence:** 4

**Review:**

This was a clearly written paper reproducing the original ELECTRA paper. The replication study authors do a great job of discussing what was difficult and easy to replication about the original paper (including the code that was released with the paper). I enjoyed reading it and thought they did an excellent job!


*Scope of reproducibility:*

This paper reproduces the GLUE scores from ELECTRA in PyTorch and on a single GPU.



*Code:*

The paper authors reimplemented ELECTRA in PyTorch (originally in TensorFlow). The authors state they will release the code, but I did not see it.


*Communication with original authors:*

It appears the replication study authors communicated with the paper authors to clear up minor details, but that the majority of questions were answered by the Github repo or the code.


*Hyperparameter Search:*

Neither the replication study nor the original paper used a hyperparameter search.

*Ablation Study:*

I don't believe the replication study performed any ablations.

I was surprised to see such similar results for ELECTRA even though the replication study did not use the WikiBooks dataset (and only used OpenWebText).

*Discussion on results:*

The replication study presented an excellent description of the reproducibility of the original paper and made clear when the results reproduced and did not reproduce. They clearly stated that some details were ambiguous, but that they were mostly able to resolve those details. The scores in the original implementation and in the new implementation mostly match.

*Recommendations for reproducibility:*

The republication study pointed out areas where the original paper did a great job of giving enough details for reproducing the work and areas where the original paper could revisit (such as the discrepancy in the WNLI tasks.

*Results beyond the paper:*

The replication study aggregated information from several sources to give additional context to the paper (such as the efficiency results). This was great to see.

*Overall organization and clarity*

* The replication study authors were very clear about where their implementation differed from the original work and included a table making it very easy to read (table 5). Thank you!

**Familiar With The Original Paper:**

I have read the original paper

**Reproducibility Summary:**

Report has summary

---

### Public Comment · ~Shaohua_Li1 · 2021-12-12
**ELECTRA vs. TinyBert**

Thanks for the hard work! My question is about the conclusion "ELECTRA is clearly outperforming all compared approaches, including BERT and TinyBERT. " However in Tables 2 and 3, it seems that TinyBERT outperforms ELECTRA? Or did I miss anything? Thanks.

---

> ### Author Response · Authors · 2021-12-13
> **Distillation vs pretrain**
>
> Thanks for your message.
>
> It depends on the criteria.
>
> If you compare by the number of parameters, yes in this table, TinyBERT outperforms, even if it's very small amount by looking at the variance of results from Electra.
>
> If you compare by amount of compute, TinyBERT requires to have already a trained BERT base model, which involves a much larger amount of compute than Electra. Since Electra-Small performances are almost similar to TinyBERT, Electra approach is more efficient than TinyBERT.
>
> Actually, distillation techniques are performed after a pretrain task (such as BERT or Electra), you could also distill a large model trained with Electra into a small model. It would be likely produce a small model outperforming TinyBERT with the same number of parameters.
>
> Regards

---

### Decision · Program_Chairs · 2021-03-31

**Decision:**

Reject

**Comment:**

While starting the code base from scratch is an interesting contribution, the reproducibility of the original paper is not carried out extensively enough.